# Reduced Reverse Cholesterol Transport Efficacy in Healthy Men with Undesirable Postprandial Triglyceride Response

**DOI:** 10.3390/biom10050810

**Published:** 2020-05-25

**Authors:** Alexandre Motte, Julie Gall, Joe-Elie Salem, Eric Dasque, Martine Lebot, Eric Frisdal, Sophie Galier, Elise F. Villard, Elodie Bouaziz-Amar, Jean-Marc Lacorte, Beny Charbit, Wilfried Le Goff, Philippe Lesnik, Maryse Guerin

**Affiliations:** 1Research Institute of Cardiovascular Disease, Metabolism and Nutrition, Faculté de Médecine - Hôpital Pitié-Salpêtrière, Sorbonne University, Inserm, UMR_S1166-ICAN, F-75013 Paris, France; alexandre.motte@inserm.fr (A.M.); julie.gall@sfr.fr (J.G.); joe-elie.salem@aphp.fr (J.-E.S.); eric.frisdal@inserm.fr (E.F.); sophie.galier@inserm.fr (S.G.); elise.villard@sanofi.com (E.F.V.); elodie.amar@aphp.fr (E.B.-A.); jean-marc.lacorte@aphp.fr (J.-M.L.); wilfried.le_goff@sorbonne-universite.fr (W.L.G.); philippe.lesnik@sorbonne-universite.fr (P.L.); 2Centre d’Investigation Clinique Paris-Est CIC-1901 Hôpital de la Pitié-Salpêtrière AP-HP, 75013 Paris, France; eric.dasque@aphp.fr (E.D.); martine.lebot@aphp.fr (M.L.); bcharbit@chu-reims.fr (B.C.)

**Keywords:** postprandial, hypertriglyceridemia, reverse cholesterol transport, high-density lipoprotein, CETP, cholesterol efflux, macrophage, triglyceride-rich lipoprotein

## Abstract

Elevation of nonfasting triglyceride (TG) levels above 1.8 g/L (2 mmol/L) is associated with increased risk of cardiovascular diseases. Exacerbated postprandial hypertriglyceridemia (PP–HTG) and metabolic context both modulate the overall efficacy of the reverse cholesterol transport (RCT) pathway, but the specific contribution of exaggerated PP–HTG on RCT efficacy remains indeterminate. Healthy male volunteers (*n* = 78) exhibiting no clinical features of metabolic disorders underwent a postprandial exploration following consumption of a typical Western meal providing 1200 kcal. Subjects were stratified according to maximal nonfasting TG levels reached after ingestion of the test meal into subjects with a desirable PP–TG response (G_Low_, TG < 1.8 g/L, *n* = 47) and subjects with an undesirable PP–TG response (G_High_, TG > 1.8 g/L, *n* = 31). The impact of the degree of PP–TG response on major steps of RCT pathway, including cholesterol efflux from human macrophages, cholesteryl ester transfer protein (CETP) activity, and hepatic high-density lipoprotein (HDL)-cholesteryl ester (CE) selective uptake, was evaluated. Cholesterol efflux from human macrophages was not significantly affected by the degree of the PP–TG response. Postprandial increase in CETP-mediated CE transfer from HDL to triglyceride-rich lipoprotein particles, and more specifically to chylomicrons, was enhanced in G_High_ vs. G_Low_. The hepatic HDL-CE delivery was reduced in subjects from G_High_ in comparison with those from G_Low_. Undesirable PP–TG response induces an overall reduction in RCT efficacy that contributes to the onset elevation of both fasting and nonfasting TG levels and to the development of cardiometabolic diseases.

## 1. Introduction

The reverse cholesterol transport (RCT) pathway, which involves the centripetal movement of free cholesterol from peripheral tissues to the liver, is recognized as the primary mechanism by which high-density lipoproteins (HDL) protect against atherosclerosis [1]. RCT represents a multistep process [2], in which the exit of intracellular cholesterol from the target cell, specifically foam cell macrophages, and its integration into HDL particles and/or lipid-poor apolipoprotein AI (apoAI), constitutes the initial step. Cellular free cholesterol efflux from human cholesterol-loaded macrophage to lipid-free or lipid-poor apoAI occurs via the ATP-Binding Cassette A1 (ABCA1) transporter, whereas free cholesterol efflux to mature HDL particles primarily involves the Scavenger Receptor-BI (SR-BI). In a second step, newly acquired free cholesterol by nascent prebeta-HDL is esterified by the Lecithin Cholesterol Acyltransferase, which allows the maturation of spherical HDL particles, small HDL3 and subsequently large HDL2. Finally, HDL-CE are delivered to the liver either directly by selective uptake of cholesteryl esters (CE) via the action of SR-BI, or indirectly following cholesteryl ester transfer protein (CETP)-mediated cholesteryl ester transfer to apoB-containing lipoproteins and the subsequent uptake of these latter particles by specific hepatic receptors.

Epidemiological studies have identified nonfasting triglycerides as a strong and independent predictor of atherosclerotic cardiovascular disease [3,4,5]. Transient elevation in circulating triglycerides represents the first and main marker of postprandial lipemia reflecting the intestinal production of chylomicrons, which are rapidly transformed into remnant particles by the action of lipoprotein lipase that reduces both triglyceride (TG) content and particle size [6]. Intravascular accumulation of liver-derived very low-density lipoprotein (VLDL) particles equally contributes to hypertriglyceridemia following meal intake [7]. Under normal metabolic context, the rapid clearance of triglyceride-rich lipoproteins (TRL) and their remnants contributes to formation of antiatherogenic HDL particles and stimulation of RCT [2], leading to the rapid CE removal from the circulation via uptake of apoB-containing lipoprotein particles via the low-density lipoprotein (LDL)-receptor pathway [8,9]. Indeed, in normolipidemic subjects, CETP-mediated CE transfer allows a massive redistribution of CE from HDL to LDL, these latter particles accounting for up to 75% of total CE transferred from HDL during postprandial phase [10].

Antiatherogenic properties of HDL particles and, more specifically, their contribution in RCT are frequently altered in dysmetabolic states characterized by a low HDL-C phenotype and a high cardiovascular risk [11]. Indeed, an overall defective RCT efficacy resulting from concomitant reduced efflux capacity and an elevated CETP activity have been reported in patients with familial hypercholesterolemia [12], type IIB hyperlipidemia [13] or hypertriglyceridemia [14]. In the meantime, overproduction and accumulation of postprandial TRL together with delayed catabolism of their remnants as observed in patients with metabolic diseases, such as metabolic syndrome, type 2 diabetes, hypertriglyceridemia or mixed hyperlipidaemia, are responsible for an exaggerated postprandial lipemia [15] and underlie the relationship between nonfasting TG and cardiovascular diseases. In such dyslipidemic patients, elevated levels of lipid poor/lipid free apoAI stimulate ABCA1-dependent cholesterol efflux from macrophages during postprandial lipemia, however, RCT efficaciousness is significantly reduced as a result of both a defective direct and a delayed indirect return of cholesterol to the liver [16], thus contributing to the development of atherosclerosis. Indeed, in hypertriglyceridemic patients accelerated CETP-mediated CE transfer from HDL preferentially targets chylomicrons and VLDL1, favoring the formation of cholesterol-enriched lipoprotein remnant particles, the well-established leading cause of cholesterol accumulation in arterial wall foam cells [17].

Alteration of RCT efficacy during postprandial state as observed in dyslipidemic subjects thus results from the combined impact of exacerbated postprandial hypertriglyceridemia and of dysmetabolic context. Therefore, no conclusive interpretation can be reached to date on the specific contribution of an exaggerated postprandial hypertriglyceridemia on RCT efficacy. The present study aims to evaluate the specific impact of an elevated postprandial hypertriglyceridemia on RCT pathway independently of the dysmetabolic context. On the basis of epidemiological studies [18,19,20], a classification for nonfasting TG concentrations has been proposed establishing a desirable postprandial TG response for subjects with nonfasting TG levels below 1.8 g/L (2 mmol/L) at any time after meal intake and an undesirable postprandial TG response in those with nonfasting TG levels above the proposed cut-off value [19]. We therefore characterized key steps of RCT, including cholesterol efflux, CETP activity and selective liver uptake of HDL-CE, during the postprandial phase in healthy men displaying no clinical features of metabolic disorders and exhibited either a desirable or an undesirable postprandial TG response.

## 2. Materials and Methods

### 2.1. Study Population

The postprandial high-density lipoprotein (HDL-PP) cohort is composed of 78 men volunteers with a fasting lipid profile within the normal range for their age. Based on clinical and biochemical fasting parameters presented in Table 1, the study population is representative of a general population of healthy nondyslipidemic men. No subject was receiving a lipid-lowering drug therapy. None had diabetes, liver, renal or thyroid disease and they were nonsmokers. Most of subjects (75/78) were nonobese, with 78% classified as normal weight (BMI < 25 kg/m^2^) and 18% as overweight (BMI > 25 kg/m² to 30 kg/m²). Considering alcohol consumption, the majority of patients were abstinent (46%) or low alcohol drinkers (51%; < 10 g/day); only two subjects were moderate drinkers (10–30 g/day). Thirty-three percent of subjects declared to have a regular physical practice defined as minimum of 30 minutes’ exercise per day. Subjects displayed either the apoE3/E3 (*n* = 45), the apoE3/E4 (*n* = 8), the apoE2/E3 (*n* = 15) or the apoE2/E4 genotype (*n* = 10). No subject exhibited the apoE2/E2 genotype.

### 2.2. Ethics

This clinical protocol was approved by the scientific ethical committee of the Pitié-Salpêtrière hospital (registration number CPP/57-11), was registered with clinical trials.gov (NCT 03109067) and conforms to the principle of the Declaration of Helsinki. Postprandial explorations were carried out at the clinical investigation center of the Pitié-Salpêtrière hospital (CIC1901 Paris-Est). Written informed consent was obtained from each patient.

### 2.3. Study Design

For each subject, a postprandial exploration was conducted as previously described [10]. Subjects were asked to abstain from alcohol and vigorous exercise for 24 h before the day of the postprandial test. A baseline blood sample was collected at 8:00 am after a 12-h overnight fasting period. A standardized breakfast of low caloric content (300 kcal; containing 12% protein, 70% carbohydrates and 18% fat) was consumed at 8:30 am, three hours before initiation of the postprandial exploration to avoid a metabolic background involving high free fatty acid levels and low insulin concentrations after a prolonged fasting overnight. Subjects consumed the test meal at 11:30 am. The test meal consisted of freshly prepared commercially available foods: instant mashed potatoes (100 g) mixed with 48 g of oil (2/3 sunflower oil and 1/3 rapeseed oil), beef steak (100 g), cheese (28 g), white bread (40 g) and apple (120 g). This meal represents a typical Western meal of 1200 kcal and consists of 14% protein, 38% carbohydrates and 48% fat, providing 66 g of fat and 142 mg of cholesterol. The subjects did not take any other meal for eight hours.

Blood samples were obtained immediately before the consumption of the test meal and two, four, six and eight hours after ingestion of the meal. Blood was collected into sterile Ethylenediaminetetraacetic acid (EDTA)-containing tubes, and plasma was separated immediately by low-speed centrifugation (2500 rpm) for 20 min at 4 °C and stored at −80 °C until use.

### 2.4. Biochemical Analyses

Biochemical measurements were performed with a calibrated autoanalyser Konelab20 (Thermo Fisher Scientific, Courtaboeuf, France) by using routine automated enzymatic methods. Triglycerides, total cholesterol, direct LDL-cholesterol, direct HDL-cholesterol and glucose were measured by using commercial kits from Thermo Fisher Scientific. Free cholesterol, phospholipids and hsCRP levels were measured by using reagents from Diasys. Apolipoprotein levels were determined using immune turbidimetric assays from Thermo Electron for apoAI and apoB levels and from Diasys for apoE, apoCII and apoCIII levels. ApoB48 levels were measured using a commercial enzyme-linked immunosorbent assay (ELISA) kit (Sobioda, Montbonnot-Saint-Martin, France). Insulin levels were determined by using human ELISA kits from Merck Millipore. Insulin resistance was evaluated using the homeostasis model assessment of insulin resistance index (HOMA-IR) using the following formula [fasting glucose (mmol/L) × fasting insulin (mUI/L)/22.5]. Bicinchoninic acid assay reagent from Pierce was utilized for total protein quantification.

### 2.5. Lipoprotein Fractionation

Chylomicrons (CM; Sf > 400) were isolated by centrifugation at 20,000 rpm for 45 min at 15 °C using a SW41 Ti rotor in a Beckman XL70 ultracentrifuge (Beckman Coulter, Villepinte, France) [21]. Subfractions of large VLDL1 (Sf 60–400) and small VLDL2 (Sf 20–60) were isolated from CM free-plasma by nonequilibrium density gradient ultracentrifugation as previously described [22]. LDL, HDL2 and HDL3 were isolated from CM free-plasma by density-gradient ultracentrifugation as described previously [22]. Lipoprotein mass was calculated as the sum of the mass of the individual components.

### 2.6. Determination of CETP Activity

Endogenous CE transfer from HDL to apoB-containing lipoproteins was assayed as previously described [23]. Briefly, CETP-mediated cholesteryl ester transfer was determined after incubation of whole plasma at 37 °C or 0 °C for 3 h in the presence of radiolabeled [^3^H]-cholesterol-HDL (25 µg HDL-CE) and iodoacetate (final concentration 1.5 mmol/L) for the inhibition of Lecithin–cholesterol acyltransferase (LCAT). After incubation, apolipoprotein B-containing lipoproteins were precipitated using the dextran sulfate-magnesium procedure. The radioactive content of the supernatant was quantified by liquid scintillation spectrometry with a 1450 Trilux Microbeta² (Perkin Elmer, Villebon-sur-Yvette, France). Endogenous plasma CETP activity (expressed as a percentage) was calculated as the amount of the label recovered in the supernatant after incubation and divided by the label present in the supernatant before incubation. The CETP-dependent CE transfer was calculated from the difference between the radioactivity transferred at 37 °C and 0 °C.

### 2.7. Cholesterol Efflux Measurements

Cholesterol efflux assays were performed as previously described [24] by using [^3^H]-cholesterol-loaded human THP-1 macrophages and cellular models representative of a specific efflux pathway, either SR-BI (Fu5AH) or ABCA1 (CHO-hABCA1 cells in which the expression of ABCA1 was induced by tetracycline). [^3^H]-cholesterol– labeled cells were incubated for 4 h at 37 °C in the presence of a 40-fold diluted plasma or fixed concentrations of isolated HDL subfractions (10 µgPL/mL for Fu5AH and 10 µgApoAI/mL for CHO-hABCA1). HDL subfractions were isolated from each subject at each time point of the postprandial exploration and stored at +4 °C until use for functional experiments which were performed within two weeks after the HDL preparation as previously recommended [25]. Fractional cholesterol efflux, expressed as a percentage, was calculated as the amount of the label recovered in the medium divided by the total label in each well (radioactivity in the medium + radioactivity in the cells) obtained after the lipid extraction from cells in a mixture of hexane isopropanol (3:2 *v/v*). The background cholesterol efflux obtained in the absence of any acceptor was subtracted from the cholesterol efflux obtained with samples. A standard plasma was tested in all experiments and was used to calculate the relative cholesterol efflux capacity of each plasma sample. All cholesterol efflux determinations were performed in triplicate for each sample.

### 2.8. In Vitro Selective Hepatic Uptake of HDL-CE

In vitro selective HDL-CE liver uptake was performed by using HepG2 cells as previously described [12]. Briefly, confluent cells were incubated in the presence of [^3^H]-CE labeled HDL (60 μg protein/mL) at 37 °C for five hours. Following incubation, the medium was removed and cells were washed with PBS and incubated in the presence of an excess of unlabeled HDL (100 μg protein/mL) for an additional period of 30 min. Then cells were washed with PBS and solubilized with 200μL of NaOH 0.2 N for 15 min before determination of the radioactive and protein content of cell lysates. Selective uptake was calculated from the known specific radioactivity of radiolabeled HDL-CE and was expressed in μgHDL-CE/μg cell protein.

### 2.9. Statistical Analyses

Variables were tested for normal distribution using the Kolmogorov–Smirnov test. The unpaired t-test was used for the intergroup comparison of normally distributed parameters, whereas the Mann–Whitney test was used for the intergroup comparisons of skewed data. Quantitative variables presented as proportions were compared using the chi square test. Repeated-measure one-way analysis of variance (ANOVA) was performed to assess changes over postprandial time course and repeated-measure two-way ANOVA was carried out to assess the changes over time among subgroups of subjects. The Bonferroni correction was applied for multiple comparisons. Postprandial variations were quantified by calculating the area under the curve (AUC) or incremental AUC (iAUC) determined before the meal intake. AUC was calculated by the trapezoidal method for the entire eight-hour period (0–8 h). The iAUC represents the increase in area in response to test meal relative to before the meal intake.

Statistical analyses were performed using the R statistical software (R Foundation for statistical computing, Vienna, Austria) computer program version Ri386 3.3.1. Results were considered statistically significant at *p* < 0.05.

## 3. Results and Discussion

### 3.1. Stratification of the Study Population According to the Degree of the Postprandial Triglyceride Response

According to epidemiological studies [18,19,20] identifying a cut-off value above 1.8 g/L (2 mmol/L) at any time after meal intake associated with an increased risk of cardiovascular diseases, we presently stratified our study population into two subgroups according to maximal nonfasting TG levels reached after the ingestion of the test meal: a subgroup with desirable PP-TG response (G_Low_, below the cut-off value of 1.8 g/L, *n* = 47) and a subgroup with undesirable PP-TG response (G_High_, above cut-off value of 1.8 g/L; *n* = 31) (Figure 1A). Of note is that the observed proportion of men (39.7%) from the HDL-PP cohort displaying an undesirable PP-TG response is entirely consistent with that previously reported for Caucasian male subjects in the large Copenhagen General Population Study, reaching 38% in men [19]. As shown in Figure 1B, in normolipidemic subjects from G_Low_, plasma TG levels peaked at 2 h after the meal intake for the majority of subjects (62%). By contrast, in normolipidemic subjects exhibiting an undesirable TG response (G_High_), maximal postprandial TG levels were attained 4 h after the meal intake for 52% of subjects, thus suggesting a prolonged rate production and/or a delayed clearance of triglyceride-rich lipoprotein (TRL) particles in subjects from G_High_ as compared to those from G_Low_. Thereafter a progressive decline in postprandial plasma TG levels occurred from 2 h to 8 h in G_Low_ or from 4 h to 8 h in G_High_, thus returning to their respective baseline TG levels eight hours after the meal intake. The two subgroups of subjects are characterized by a significant distinct magnitude of the postprandial TG response after consumption of a typical Western meal as shown by a significant 2.4-fold (*p* < 0.0001) higher iAUC of TG in G_High_ compared with G_Low_ (Figure 1C).

As presented in Table 2, clinical and biochemical fasting parameters of subjects belonging to the G_High_ subgroup were within the upper range of normal values, whereas those of G_Low_ were within the lower range of normal values. Specifically, the fasting plasma lipid profile of normolipidemic subjects with an undesirable postprandial TG response trends towards a proatherogenic triad characterized by elevated levels of total cholesterol (*p* < 0.02), LDL-Cholesterol (*p* < 0.05) and triglycerides (*p* < 0.0001) and reduced levels of HDL-Cholesterol (*p* < 0.02). Equally, it is relevant to consider that fasting remnant lipoprotein-cholesterol levels were significantly higher (*p* < 0.0001) in G_High_ compared with G_Low_, thus indicating higher fasting levels of residual triglyceride-rich lipoprotein remnant particles in subjects from G_High_. In addition, a significant elevation in body mass index and, more specifically, in abdominal obesity as well as in glycated hemoglobin (HbA1c) plasma levels were observed in the G_High_ as compared to G_Low_. Interestingly, 28% of subjects from G_Low_ and 42% of subjects from G_High_ meet the criteria of prediabetes as defined by the American Diabetes Association [26], which include (a) an impaired fasting glucose with fasting plasma glucose between 5.6 mmol/L to 6.9 mmol/L and/or (b) an impaired glucose tolerance defined as glucose levels between 7.8 mmol/L to 11 mmol/L after a 75 gr oral glucose tolerance test and/or (c) HbA1c plasma levels of 5.7 to 6.4%. Those patients equally displayed one or two additional criteria characteristic of prediabetes in asymptomatic adults including abdominal obesity (waist circumference ≥ 102 cm) and a low HDL-C phenotype (≤40 mg/dL); however the higher relative proportion of prediabetic subjects presently observed among G_High_ as compared to G_Low_ did not reach the statistical significance (*p* = 0.19).

### 3.2. Quantitative and Qualitative Features of Plasma Lipoprotein Subspecies According to the Degree of the Postprandial Triglyceride Response

As shown in Figure 2, postprandial hypertriglyceridemia following consumption of a typical Western solid mixed meal predominantly reflected postprandial changes in triglyceride-rich lipoprotein particles including chylomicrons and VLDL1. In comparison with normolipidemic subjects from G_Low_, individuals exhibiting an undesirable postprandial TG response were characterized by a marked postprandial elevation in circulating levels of both CM-TG (2.4-Fold; *p* < 0.0001; Figure 2A) and VLDL1-TG (2.3-Fold; *p* < 0.0001, Figure 2B), whereas those of VLDL2-TG were not significantly increased (*p* = 0.36, Figure 2C).

Similarly, postprandial elevations in CM-apoB48 and VLDL1-apoB100 levels, reflecting increased numbers of circulating triglyceride-rich lipoprotein particles of intestinal or hepatic origin, respectively, during the postprandial phase, were significantly higher in subjects from G_High_ in comparison with subjects from G_Low_ (1.6-Fold, *p* = 0.014 and 1.4-Fold, *p* = 0.010 for postprandial change in CM-apoB48 and VLDL1-apoB100, respectively). Note that in subjects from both G_High_ and G_Low_, plasma lipid parameters as well as circulating number of individual TRL subspecies return approximately to their respective baseline levels within eight hours.

More strikingly, we observed that postprandial CM and VLDL1 particles isolated from subjects belonging to the G_High_ subgroup were significantly enriched in TG as shown by the elevation of the TG/apoB ratio in CM (+69%; *p* < 0.001) and VLDL1 (+20%; *p* < 0.01) isolated four hours after the meal intake from G_High_ as compared to those isolated from G_Low._ Taken together, these observations support the contention that the undesirable postprandial TG response might result from a reduced LPL-mediated TG hydrolysis and a delayed clearance of apoB-containing TRL particles. In this context, it is relevant to note that fasting apoB48 levels were significantly higher in G_High_ compared with G_Low_, thus indicating higher fasting levels of residual apoB48-containing remnant particles of intestinal origin in subjects from G_High_ (Table 2). In addition, plasma levels of apoCIII were equally significantly increased by +37% (*p* = 0.0008) in G_High_ in comparison with G_Low._ More specifically, we presently observed significant higher circulating levels of CM-apoCIII, VLDL1-apoCIII and, to a lesser extent, of VLDL2-apoCIII in subjects from G_High_ as compared to those from G_Low_ (Figure 3A,B). VLDL2-apoCIII accounted for approximately 20% of total apoCIII bound to TRL particles **(**Figure 3C). These latter observations are in good agreement with earlier studies showing a higher proportion of apoCIII bound to TRL particles and their remnants in hypertriglyceridemic subjects [27]. It is well established that apoCIII acts as an inhibitor of lipoprotein lipase and more specifically inhibits the lipoprotein lipase-mediated lipolysis of TG-rich lipoproteins [28]. ApoCIII equally impairs the hepatic uptake of TRLs by remnant receptors including the LDL-R (low-density lipoprotein-receptor) and the LSR (lipolysis-stimulated receptor) [29]. Thus, the undesirable postprandial hypertriglyceridemia presently observed in subjects from G_High_ might primarily result from the combination of increased apoCIII concentrations, reduced activities of LPL and hepatic remnant receptors, together with competition of intestinal and hepatic-derived lipoproteins for common removal pathways thus resulting in a delayed catabolism of TRLs and their remnants.

Earlier studies have demonstrated that apoE isoforms influence metabolism of postprandial lipoprotein particles [30]. In particular, while apoE2/E2 carriers displayed a delayed clearance of TRL particles underlying the establishment of dysbetalipoproteinemia [31], apoE4/E3 subjects are characterized by an accelerated postprandial clearance of chylomicron-remnants as compared to E3/3 individuals and up to twice fast in comparison with E3/2 individuals [30]. In the context of the present study, it is relevant to consider that no apoE2/E2 carriers were included and that no significant difference (*p* = 0.60) in the relative proportions of each apoE genotype between the two subgroups of subjects was observed.

### 3.3. Cellular Free Cholesterol Efflux Capacity According to the Degree of the Postprandial Triglyceride Response

Postprandial state was without significant impact on the capacity of whole plasma to mediate cholesterol efflux from human cholesterol-loaded THP-1 macrophages (Figure 4A) and was not influenced by the degree of the postprandial TG response. Similarly, ABCA1-dependent plasma efflux capacity remained unchanged throughout the postprandial phase (Figure 4B). A slight but significant elevation (+8%; *p* < 0.05) in SR-BI dependent efflux was observed during the late postprandial phase, specifically in subjects from G_Low_ group (0.99 ± 0.26 and 1.07 ± 0.25, before and at 8 h after the meal intake, respectively) whereas no significant variation in plasma efflux capacity via SR-BI was detected during the postprandial phase in G_High_ (Figure 4D). In addition, no significant impact of the postprandial state was detected on the capacity of HDL subspecies, large HDL2 or small HDL3, to mediate cellular free cholesterol efflux via either ABCA1 (Figure 4C) or SR-BI (Figure 4E,F). However, an overall significant higher mean postprandial ABCA1-dependent plasma efflux capacity (+10%; *p* =0.0125) in G_High_ as compared to G_Low_ was observed (1.00 ± 0.35 and 0.90 ± 0.36 in G_High_ and G_Low_, respectively), suggesting the presence of elevated circulating levels of preferential acceptors of cellular cholesterol via ABCA1, namely lipid-poor prebeta-HDL particles, in subjects with higher levels of TG (Figure 4B). Indeed, Stock et al. have previously reported elevated levels of prebeta-HDL particles in various dyslipidemic states including hypertriglyceridemia [32]. Equally, the mean capacity of postprandial large HDL2 particles for SR-BI mediated cholesterol efflux appeared to be significantly increased (+7.7%; *p* = 0.0197) in subjects from G_Low_ in comparison with their counterparts from G_High_ (5.15 ± 1.16 and 4.78 ± 1.41, in G_Low_ and G_High_, respectively). These observations are entirely consistent with earlier studies demonstrating that HDL particles isolated from patients displaying metabolic disorders and more specifically hypertriglyceridemia, display a reduced capacity to mediate SR-BI–dependent efflux [33,34], whereas ABCA1-dependent efflux has been shown uniformly accelerated among dyslipidemic patients [35,36,37]. Quantitative and qualitative features of postprandial HDL subfractions are presented in Table 3**.** Note that a reduction in the CE/TG ratio in HDL particles, resulting from a reduction in TG content, was observed in subjects from G_High_ as compared to G_Low_, before the meal intake and during the postprandial lipemia.

Taken together, our observations indicate that each cholesterol efflux pathway contributes differentially to the overall plasma efflux capacity from human macrophage in normolipidemic subjects exhibiting either a desirable or an undesirable postprandial TG response. In addition, despite the presence of less effective HDL particles for SR-BI dependent efflux, the overall capacity of plasma to remove cholesterol from human THP-1 macrophage is maintained in normolipidemic subjects characterized by elevated postprandial triglyceride levels.

### 3.4. Endogenous CETP Activity According to the Degree of the Postprandial Triglyceride Response

In good agreement with earlier studies [8,10], plasma CETP activity, expressed as the percentage of CE transferred from HDL to apoB-containing lipoproteins, was significantly increased following meal intake in both subgroups of normolipidemic subjects, G_Low_ and G_High_ (Figure 5A). Interestingly, whereas a slight but significant increase (+10.2%; *p* < 0.05) in postprandial plasma CETP activity was detected 4 h after the meal intake in subjects from G_Low_, a more pronounced postprandial elevation in CETP-dependent CE transfer occurred at 2 h (+13%; *p* < 0.01), 4 h (+23.7%; *p* < 0.001) and at 6 h (+17.5%; *p* < 0.001) in subjects from G_High_. These observations indicate that the undesirable postprandial TG response was characterized by an accelerated CE transfer from HDL to apoB-containing lipoproteins (AUC for CETP, +17.2%; *p* = 0.006 in G_High_ versus G_Low)._ As expected, CETP-dependent CE transfer was positively correlated with both circulating levels of LDL-C (*r* = 0.462; *p* < 0.0001) and TG (*r* = 0.378; *p* < 0.0001) and was inversely correlated with those of HDL-C (*r* = −0.579; *p* < 0.0001). Indeed, it is well established that endogenous CETP activity is closely linked to circulating levels of each individual lipoprotein subspecies which represent strong modulators of CETP-mediated neutral lipid transfer exchange between lipoprotein particles [38]. Interestingly, in subjects from G_Low_ subgroup, no significant relationship was observed between the postprandial CETP activity and absolute change in CM-TG levels whereas significant positive correlations were observed with those of large VLDL1-TG and small VLDL2-TG levels (Figure 5B–D). These latter observations are in good agreement with earlier studies [10], showing that chylomicrons do not represent major cholesteryl ester acceptors of CE during the postprandial phase in subjects with a desirable postprandial TG response. By contrast, in subjects displaying an undesirable postprandial TG response, an enhanced CETP-mediated CE transfer to chylomicron and VLDL1 together with a reduced transfer to VLDL2 were observed, as shown by the significant positive relationship between the absolute change in CM-TG or in VLDL1-TG and the postprandial CETP activity (Figure 5B,C) and the absence of significant correlation between the postprandial change in VLDL2-TG and CETP activity (Figure 5D). Note that such an enhanced cholesteryl ester transfer from HDL to triglyceride-rich lipoprotein subspecies has been previously reported in patients presenting primary hypertriglyceridemia [39,40] or during the postprandial phase in subjects with mixed hyperlipidemia [41].

### 3.5. Selective HDL-CE Uptake to Hepatic Cells According to the Degree of the Postprandial Triglyceride Response

The capacity of HDL particles isolated from subjects exhibiting a desirable TG response to deliver CE to hepatic cells remained unchanged throughout the postprandial phase, whereas those isolated from subjects characterized by an undesirable TG response showed a significant reduction of their capacity to deliver CE to hepatic cells at 4 h (−5.9%; *p* < 0.05), 6 h (−6.4%; *p* < 0.05) and 8 h (−5.7%; *p* < 0.05) after the meal consumption as compared to before the meal intake (Figure 6). In consequence, postprandial HDL isolated from G_High_ subgroup displayed a significant reduced capacity to deliver CE to hepatic cells as compared to their equivalent counterpart isolated from subjects from G_Low_ (−19.5%; *p* < 0.01 and −15.4%; *p* < 0.05, at 6 h and 8 h, respectively). These observations indicate that the undesirable PP-TG response was associated with an overall significant reduction in the capacity of postprandial HDL particles to deliver CE to hepatic cells (AUC for HDL-CE uptake −11.1%; *p* = 0.0018, in subject from G_High_ versus G_Low_). Interestingly, the capacity of HDL particles to deliver CE to hepatic cells positively correlated with the HDL-CE/TG ratio (*r* = 0.80; *p* = 0.005), whereas an inverse relationship was observed with the HDL-apoCIII content (*r* = −0.63; *p* = 0.003). Those observations are entirely consistent with earlier studies showing that the CETP-mediated reduction in the CE/TG ratio in HDL particles due to a TG enrichment, as observed in subjects from G_High_ as compared to G_Low_ or during postprandial lipemia (Table 3) is associated with a significant decrease in the SR-BI-mediated CE liver uptake while the HDL-TG depletion enhanced the HDL-CE selective hepatic uptake [42]. Equally, earlier studies have demonstrated that ApoCIII can bind to hepatic SR-BI and inhibit the selective uptake of cholesterol from HDL [43].

## 4. Conclusions

Our observations highlight that the degree of the postprandial hypertriglyceridemia in healthy male subjects without any marked clinical features of metabolic disorder, differentially modulates key steps of the reverse cholesterol transport pathway. Whereas cellular free cholesterol efflux from macrophages was not affected by the degree of the postprandial hypertriglyceridemia, the CETP-mediated CE transfer from HDL to TRL particles was enhanced and the hepatic HDL-CE delivery was reduced in subjects exhibiting an undesirable postprandial TG response in comparison with those displaying a desirable postprandial TG response (Figure 7). Such an overall reduced reverse cholesterol transport efficacy thus participates to the onset of the postprandial HTG and therefore to the early development of cardiometabolic diseases in normolipidemic healthy males.

Finally, our study underlines that healthy men subjects displaying fasting TG levels within the normal range but exhibiting an undesirable transient postprandial TG response share major common underlying mechanisms observed during metabolic hypertriglyceridemia, characterized by elevated fasting TG levels and contributing to the development of atherosclerosis. An abnormal postprandial TG response above 1.8 g/L (2 mmol/L), as proposed by the expert panel for postprandial HTG [19,20], should be more widely use in clinical practice for the early identification of subjects at high cardiovascular risk prior the appearance of any clinical features of metabolic disorder. Lifestyle interventions, targeting lipid levels or body mass index mildly above normal, with dietary changes and increase physical activity represent the cornerstone of the clinical management in the early stage of development of metabolic disorders, prior to the introduction of lipid-lowering drugs, in order to prevent further development of dyslipidemia or diabetes and their consequences [44]. However, to date, significant difficulties remain in identifying individuals at high risk of premature atherosclerosis among those displaying no marked clinical features of metabolic disorders. Undesirable nonfasting TG levels above 1.8g/L (2 mmol/L), which reflect not only an abnormal postprandial TRL metabolism but also a reduced RCT efficacy that underlies development of premature atherosclerosis, might thus represent a useful tool for the early identification of those subjects.

## Figures and Tables

**Figure 1 biomolecules-10-00810-f001:**
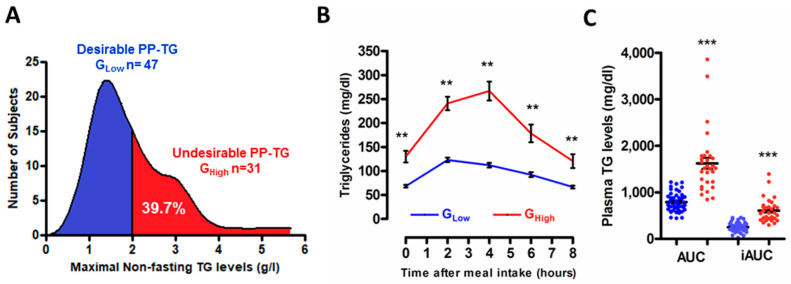
Postprandial triglyceride (TG) response following consumption of a typical solid mixed meal. Distribution of nonfasting TG levels in normolipidemic men from the postprandial high-density lipoprotein (HDL-PP) cohort (*n* = 78). Subjects were divided into two subgroups according maximum nonfasting TG levels below or above the cut-off value of 1.8 g/L (2 mmol/L) identifying subjects with a desirable postprandial TG response (G_Low_, below 1.8 g/L; *n* = 47) or an undesirable postprandial TG response (G_High_, above 1.8 g/L; *n* = 31) (**A**). Postprandial time course of plasma triglyceride levels in response to the ingestion of the test meal in subjects from G_Low_ (blue curve) and G_High_ (red curve) (**B**). Area under the curve (AUC) and incremental AUC (iAUC) in subjects from G_Low_ (blue dots) and G_High_ (red dots) (**C**). Values are mean ± SEM. *** *p* < 0.0001 and ** *p* < 0.001 versus G_Low_.

**Figure 2 biomolecules-10-00810-f002:**
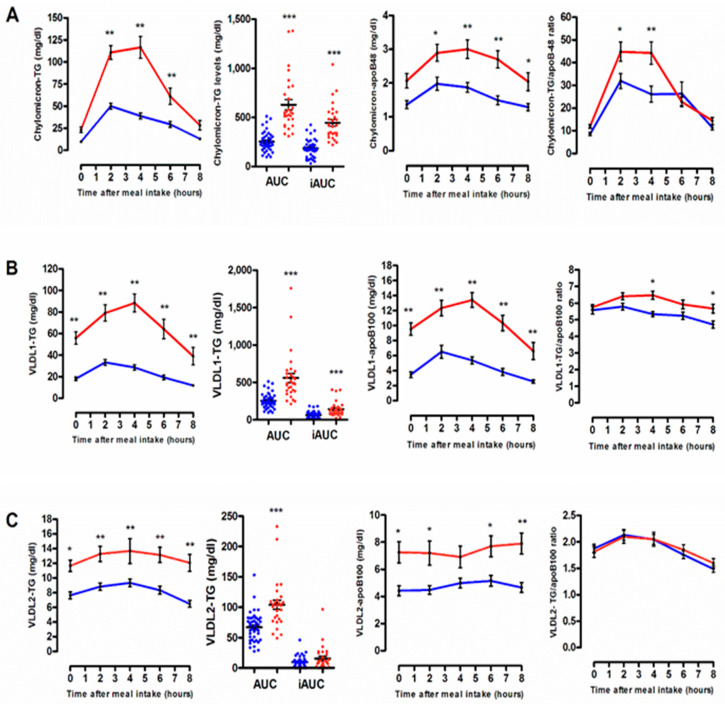
Postprandial time course of triglyceride-rich lipoprotein subfractions, chylomicrons (**A**), large very-low-density lipoprotein-1 (VLDL1) (**B**) and small VLDL2 (**C**) isolated from subjects from G_Low_ (blue curve) and G_High_ (red curve). In all sections are represented plasma levels of TRL-TG, area under the curve (AUC) and incremental AUC (iAUC) in subjects from G_Low_ (blue dots) and G_High_ (red dots), plasma levels of triglyceride-rich lipoprotein (TRL)-apoB and TRL-TG/apoB ratio. Values are mean ± SEM. *** *p* <0.0001, ** *p* <0.001 and * *p* <0.05 versus G_Low_.

**Figure 3 biomolecules-10-00810-f003:**
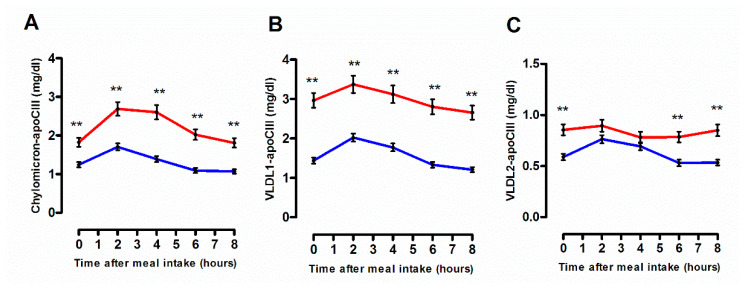
Postprandial plasma levels of chylomicron-apoCIII (**A**), large VLDL1-apoCIII (**B**) and small VLDL2-apoCIII (**C**) from G_Low_ (blue curve) and G_High_ (red curve). Values are mean ± SEM. ** *p* < 0.001 versus G_Low_.

**Figure 4 biomolecules-10-00810-f004:**
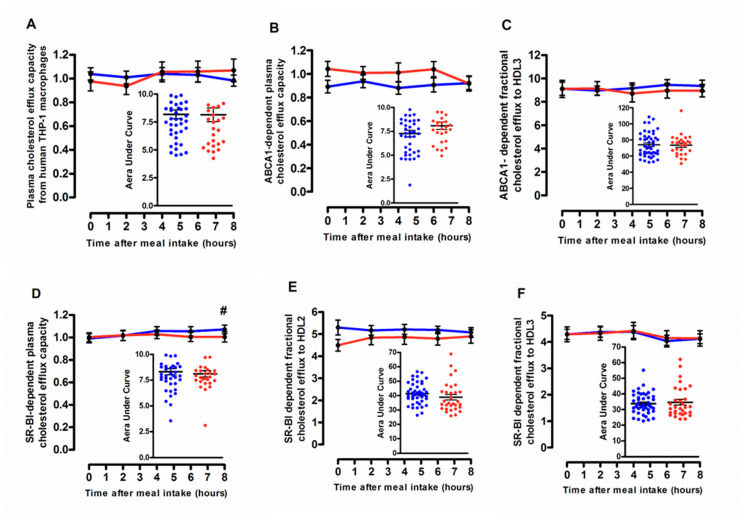
Cholesterol efflux capacity of plasma or isolated high-density lipoprotein (HDL) particles in subjects from G_Low_ (blue curve) and G_High_ (red curve). Line plots showing the capacity of 40-fold diluted plasma to mediate cellular free cholesterol efflux from human cholesterol-loaded THP-1 macrophages (**A**), via ABCA1 (**B**) and via Scavenger Receptor-BI (SR-BI) (**D**). ABCA1-dependent efflux capacity of isolated postprandial HDL3 (**C**); SR-BI dependent efflux capacity of isolated postprandial HDL2 (**E**) and HDL3 (**F**). Inserts represent area under the curve (AUC) in subjects from G_Low_ (blue dots) and G_High_ (red dots). Values are mean ± SEM. # *p* < 0.05 versus before meal intake.

**Figure 5 biomolecules-10-00810-f005:**
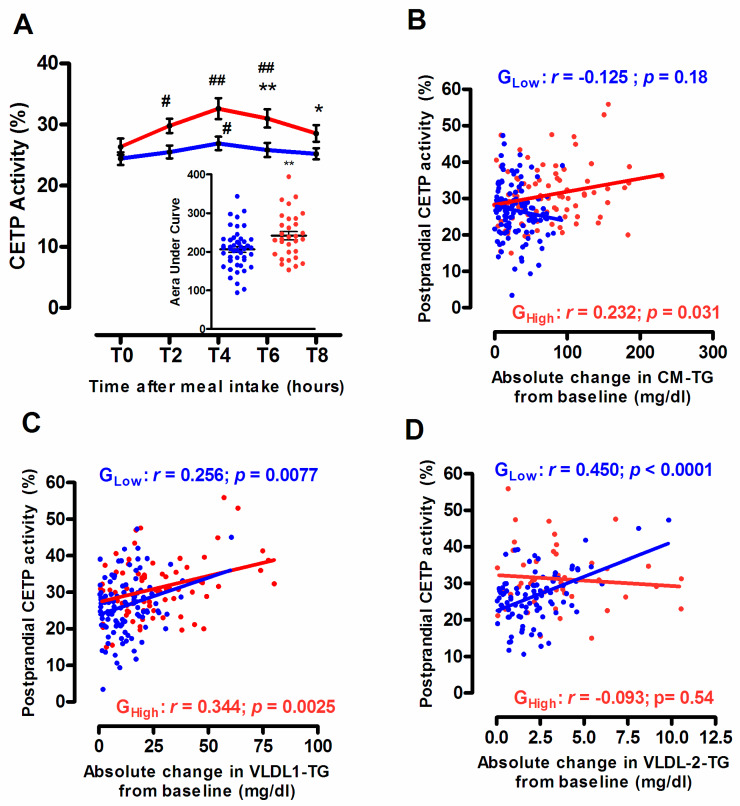
Postprandial time course of endogenous cholesteryl ester transfer (CETP) activity in subjects from G_Low_ (blue curve) and G_High_ (red curve). Insert represents the area under the curve (AUC) in subjects from G_Low_ (blue dots) and G_High_ (red dots) (**A**). Relationship between absolute changes in postprandial CM-TG (**B**), large VLDL1-TG (**C**), small VLDL2-TG (**D**) levels determined at 2 h, 4 h and 6 h after the meal intake and CETP activity in subjects from G_Low_ (blue line) and G_High_ (red line). Values are mean ± SEM. * *p* < 0.05 and ** *p* < 0.01 versus G_Low_. # *p* < 0.05 and ## *p* < 0.001 versus before the meal intake.

**Figure 6 biomolecules-10-00810-f006:**
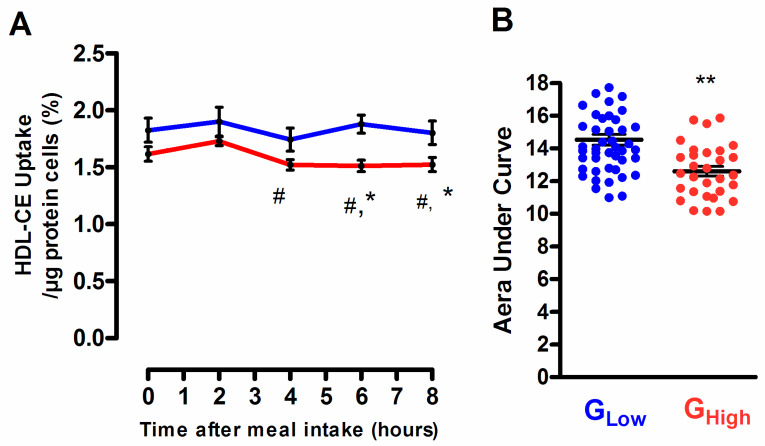
Postprandial time course of the in vitro capacity of postprandial HDL particles isolated from subjects with a desirable postprandial TG response, G_Low_ (blue curve) and with an undesirable postprandial TG response, G_High_ (red curve) to deliver CE to HepG2 cells (**A**). Area under the curve (AUC) in subjects from G_Low_ (blue dots) and G_High_ (red dots) (**B**). Values are mean ± SEM. ** *p* < 0.002 and * *p* < 0.05 versus G_Low_. # *p* < 0.05 versus before meal intake.

**Figure 7 biomolecules-10-00810-f007:**
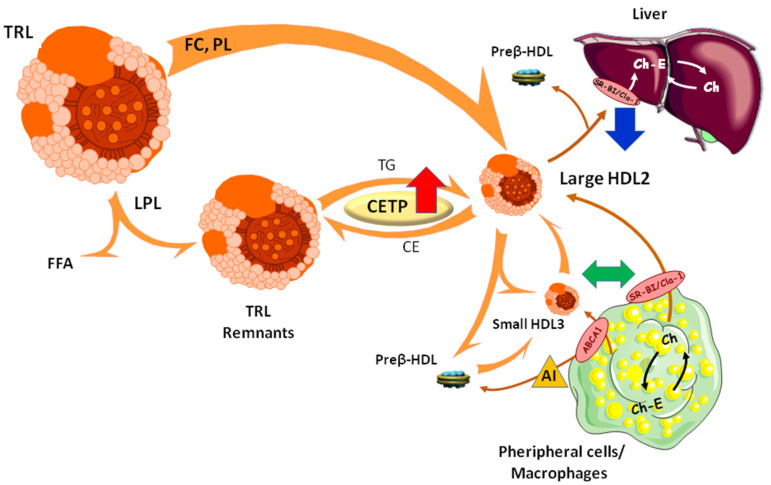
Schematic representation of the reverse cholesterol transport pathway and intravascular remodeling of lipoprotein particles during the postprandial phase. Large arrows indicate steps of the reverse cholesterol transport pathway that are enhanced (red arrow), reduced (blue arrow) or unchanged (green arrow) in normolipidemic male subjects exhibiting an undesirable postprandial TG response (G_High_) compared to those displaying a desirable postprandial TG response (G_Low_).

**Table 1 biomolecules-10-00810-t001:** Clinical and biological characteristics of the study population.

Variables	Male Volunteers from the HDL-PP Cohort (*n* = 78)
Age, y	30.1 ± 11.7
BMI, kg/m²	23.1 ± 2.7
Waist circumference, cm	83.8 ± 9.6
Hip circumference, cm	92.2 ± 8.0
Waist to Hip ratio	0.91 ± 0.07
Systolic blood pressure, mmHg	119.6 ± 9.0
Diastolic blood pressure, mmHg	77.2 ± 9.0
Fasting blood glucose, mmol/L	4.66 ± 0.45
HbA1c, %	5.32 ± 0.34
Insulin, mU/L	3.61 (1.61–6.34)
HOMA-IR	0.70 (0.31–1.22)
Creatinine, µmol/L	87.9 ± 9.23
Creatinine Clearance, mL/min	112.2 (101.9–122.7)
ASAT, UI/L	25.9 ± 5.1
ALAT, UI/L	23.3 ± 9.6
TSH, mUI/L	1.73 ± 0.70
hsCRP, mg/L	0.58 (0.10–1.50)
Triglycerides, mg/dL	64.1 (44.8–88.3)
Total cholesterol, mg/dL	170.7 ± 32.6
LDL-Cholesterol, mg/dL	102.6 ± 27.6
HDL-Cholesterol, mg/dL	49.5 ± 11.9
Remnant-Lipoprotein Cholesterol, mg/dL	15.9 (12.3–22.8)
ApoAI, mg/dL	127.2 ± 19.7
ApoAII, mg/dL	35.2 ± 6.12
ApoB-48, mg/dL	0.78 (0.54–1.21)
ApoB-100, mg/dL	73.7 ± 24.2
ApoE, mg/dL	3.20 ± 1.33
ApoCII, mg/dL	4.02 ± 2.40
ApoCIII, mg/dL	7.30 ± 2.95

Values are mean ± SD or median (interquartile range).

**Table 2 biomolecules-10-00810-t002:** Clinical characteristics and fasting lipid parameters of subjects with desirable and undesirable postprandial triglyceride response.

	Male Volunteers from the HDL-PP Cohort		
Variables	Desirable PP-TG response (*n* = 47)	Undesirable PP-TG response (*n* = 31)	*p*-Value
Age, y	28.3 ± 10.7	32.9 ± 12.7	0.0879
BMI, kg/m²	22.5 ± 2.2	24.1 ± 3.2	0.0207
Waist circumference, cm	81.5 ± 7.2	87.3 ± 11.7	0.0189
Waist to Hip ratio	0.89 ± 0.05	0.94 ± 0.08	0.0120
Fasting blood glucose, mmol/L	4.62 ± 0.43	4.73 ± 0.49	0.3181
HbA1c, %	5.25 ± 0.36	5.43 ± 0.28	0.0184
Insulin, mU/L	3.34 (1.00–5.77)	4.02 (1.77–6.85)	0.1443
HOMA-IR	0.68 (0.23–1.16)	0.80 (0.41–1.51)	0.1603
Lipids, mg/dL			
Triglycerides	55.6 (42.1–67.3)	87.9 (62.8–101.4)	<0.0001
Total cholesterol	163.3 ± 30.5	181.9 ± 33.0	0.0125
LDL-Cholesterol	97.5 ± 27.4	110.3 ± 26.6	0.0453
HDL-Cholesterol	52.0 ± 11.0	45.6 ± 12.5	0.0197
Remnant-Lipoprotein Cholesterol	13.1 (11.1–16.2)	24.3 (18.0–28.3)	<0.0001
Apolipoproteins, mg/dL			
AI	126.7 ± 18.5	127.9 ± 21.8	0.7911
AII	34.3 ± 5.3	36.6 ± 7.1	0.1060
B-48	0.69 (0.46–0.95)	1.06 (0.75–1.90)	0.0109
B-100	65.3 ± 19.1	86.3 ± 25.8	<0.0001
E	3.03 ± 0.98	3.46 ± 1.71	0.2167
CII	3.55 ± 1.79	4.65 ± 2.95	0.1785
CIII	6.41 ± 2.30	8.79 ± 3.17	0.0008

Parameters were determined on fasting samples obtained after an overnight fast, at 8:00 am. Values are mean ± SD or median (interquartile range).

**Table 3 biomolecules-10-00810-t003:** Plasma levels and chemical composition of postprandial HDL subfractions in subjects with a desirable and an undesirable postprandial triglyceride response.

			Chemical Component, % Weight					
	Hours	Total Mass	FC	CE	TG	PL	Protein	CE/TG
HDL2, d = 1.063–1.125 g/mL								
Desirable PP-TG response (G_Low_)	0	153.2 ± 6.1	6.0 ± 0.2	23.7 ± 0.5	4.0 ± 0.2	21.8 ± 0.3	44.4 ± 0.5	5.97
	2	164.2 ± 8.0	5.9 ± 0.2	22.8 ± 0.6	4.3 ± 0.2	22.2 ± 0.3	44.8 ± 0.4	5.29
	4	173.3 ± 7.1 #	5.9 ± 0.2	22.5 ± 0.6	4.7 ± 0.2 ##	22.5 ± 0.3	44.4 ± 0.5	4.79 ##
	6	179.6 ± 5.7 ##	6.0 ± 0.2	23.1 ± 0.7	4.5 ± 0.2 #	22.4 ± 0.3	44.0 ± 0.5	5.10
	8	183.6 ± 6.8 ##	5.9 ± 0.1	24.0 ± 0.5	3.8 ± 0.2	22.4 ± 0.2	43.9 ± 0.4	6.27
Undesirable PP-TG response (G_High_)	0	150.6 ± 7.0	5.7 ± 0.2	22.2 ± 0.6	5.2 ± 0.3 *	21.3 ± 0.3	45.6 ± 0.4	4.27 *
	2	155.8 ± 7.1	4.9 ± 0.2	22.7 ± 0.5	5.7 ± 0.3 *	21.8 ± 0.3	44.8 ± 0.4	4.01 *
	4	165.0 ± 7.8 #	5.1 ± 0.2	20.9 ± 0.5 #	6.8 ± 0.3 ##, **	21.8 ± 0.2	45.4 ± 0.4	3.09 ##, *
	6	163.2 ± 6.5 #	5.1 ± 0.2	20.9 ± 0.6 #	6.1 ± 0.3 ##, **	22.4 ± 0.2	45.4 ± 0.5	3.40 ##, *
	8	168.9 ± 6.9 ##	5.2 ± 0.2	21.5 ± 0.6 *	5.2 ± 0.3 *	22.2 ± 0.3	45.8 ± 0.4	4.12 *
HDL3, d = 1.125–1.21 g/mL								
Desirable PP-TG response (G_Low_)	0	140.4 ± 3.8	3.0 ± 0.1	16.1 ± 0.3	3.6 ±0.1	18.3 ± 0.4	58.9 ± 0.5	4.47
	2	136.0 ± 2.7	3.1 ± 0.2	14.9 ± 0.4 #	3.8 ± 0.2	18.8 ± 0.3	59.3 ± 0.5	3.89 #
	4	134.7 ± 3.2	3.2 ± 0.2	14.0 ± 0.4 ##	4.1 ± 0.2 #	18.9 ± 0.3	59.7 ± 0.5	3.41 ##
	6	134.0 ± 3.0	3.2 ± 0.1	14.3 ± 0.4 ##	4.3 ± 0.2 ##	18.9 ± 0.3	59.2 ± 0.5	3.31 ##
	8	140.1 ± 3.7	3.1 ± 0.1	15.2 ± 0.4	3.9 ± 0.2	19.3 ± 0.3	58.5 ± 0.5	3.91 #
Undesirable PP-TG response (G_High_)	0	153.7 ± 4.1	3.1 ± 0.2	16.1 ± 0.6	4.3 ± 0.2	17.8 ± 0.3	58.4 ± 0.5	3.71
	2	145.1 ± 4.5	2.9 ± 0.1	15.2 ± 0.4	4.8 ± 0.2 *	18.2 ± 0.4	58.9 ± 0.5	3.16 #
	4	145.7 ± 4.0	2.9 ± 0.1	13.5 ± 0.6 ##	5.2 ± 0.2 #, *	18.0 ± 0.5	60.4 ± 1.0	2.58 ##
	6	140.8 ± 3.3	2.8 ± 0.1	13.9 ± 0.4 ##	5.2 ± 0.2 #, *	18.9 ± 0.4	59.2 ±0.5	2.69 ##
	8	142.3 ± 3.6	3.1 ± 0.1	14.2 ± 0.5	4.7 ± 0.2, *	18.8 ± 0.4	59.2 ± 0.6	2.98 ##

Plasma levels and chemical composition of HDL subfractions were determined throughout the postprandial phase, before meal intake (0 h) and 2 h, 4 h, 6 h and 8 h after ingestion of the meal, in subjects with a desirable (G_Low_) or an undesirable (G_High_) postprandial TG response. Values are mean ± SEM. **p* < 0.05, ** *p* <0.001 vs G_Low_. # *p* <0.05 and ## *p* < 0.001 vs before the meal intake.

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
