# Peer review of "Reduced Reverse Cholesterol Transport Efficacy in Healthy Men with Undesirable Postprandial Triglyceride Response"

_biomolecules, 2020, doi:10.3390/biom10050810_

Round 1
Reviewer 1 Report
This is an interesting and well-performed study. The authors observed that the extent of postprandial hypertriglyceridemia in healthy men is positively associated with CETP activity and decreased HDL-CE uptake by the liver.
- How were cholesterol efflux experiments performed? Was isolated HDL from each subject used ore were experiments performed with pooled samples? Please clarify.
- Clarifying some underlying mechanisms would improve this manuscript. HDL associated apoC-III is well known to delay HDL uptake by the liver. The authors should assess whether apoC-III (or other components) of HDL are linked to the decreased uptake by the liver.
- How was isolated HDL stored after isolation for further experiments? This is critical given that incorrect storage of HDL markedly affects HDL functionality (PMID: 28893842) – please clarify.
Author Response
Responses to R1
Comment: This is an interesting and well-performed study. The authors observed that the extent of postprandial hypertriglyceridemia in healthy men is positively associated with CETP activity and decreased HDL-CE uptake by the liver.
Reponse: We thank this Reviewer for his/her helpful and constructive comments which have allowed us to significantly improve the impact and the clarity of our manuscript
Comment: How were cholesterol efflux experiments performed? Was isolated HDL from each subject used ore were experiments performed with pooled samples? Please clarify.
Response: HDL subfractions were isolated from each subject at each time point of the postprandial exploration.
This point is now more clearly indicated in section 2.7.
Comment: Clarifying some underlying mechanisms would improve this manuscript. HDL associated apoC-III is well known to delay HDL uptake by the liver. The authors should assess whether apoC-III (or other components) of HDL are linked to the decreased uptake by the liver.
Response: The capacity of HDL particles to deliver CE to hepatic cells positively correlated with HDL-CE/TG ratio (r=-0.80; p=0.005), whereas an inverse relationship was observed with HDL-apoCIII content (r=-0.63; p=0.003).
Section 3.5 has been completed as follows “undesirable PP-TG response was associated with an overall significant reduction in the capacity of postprandial HDL particles to deliver CE to hepatic cells (AUC for HDL-CE uptake -11.1%; p=0.0018, in subject from GHigh versus GLow). Interestingly, the capacity of HDL particles to deliver CE to hepatic cells positively correlated with HDL-CE/TG ratio (r=0.80; p=0.005), whereas an inverse relationship was observed with HDL-apoCIII content (r=-0.63; p=0.003). Those observations are entirely consistent with earlier studies showing that CETP-mediated reduction in CE/TG ratio in HDL particles due to TG enrichment, as observed in subjects from GHigh as compared to GLow or during postprandial lipemia (Table 3) is associated with a significant decrease in SR-BI-mediated CE liver uptake while HDL-TG depletion enhanced HDL-CE selective hepatic uptake [42]. Equally, earlier studies have demonstrated that ApoCIII can bind to hepatic SR-BI and inhibit selective uptake of cholesterol from HDL [43].
Comment: How was isolated HDL stored after isolation for further experiments? This is critical given that incorrect storage of HDL markedly affects HDL functionality (PMID: 28893842) – please clarify.
Response: We agree that this point is of critical importance. Isolated HDL subfractions were stored at +4°C until use in functional experiments which were performed within two weeks after HDL preparation as previously recommended [PMID28893842].
This point is now clearly indicated in section 2.7.
Reviewer 2 Report
Reduced Reverse Cholesterol Transport efficacy in healthy men with undesirable postprandial
triglyceride response
This work investigate some of the functions of HDL in response to postprandial elevated triglyceride (TG) levels. The overall outcome is consistent with several previous studies on related topic. In this work the authors have done a systematic evaluation of some of the key functions of HDL and correlate it with TG levels. The study design is clearly explained and the data is satisfactory and supportive of their conclusion. These are some minor points that needs clarification for better understanding.
- What is the reason for selectively choosing male subjects?
- In section 2.3, under study design, is it right to assume that after test meal, the subjects were not taking any other meal for 8h? If so, please add this detail in this section.
- Table 2 and 3, please give the time point that correspond to these values.
- One of the conclusion from Table 2 and Figure 2 is elevated TG levels in CM and VLDL1 in Ghigh subgroup and the authors correlate this with elevated apoCIII levels. Do they have any data that show elevated apo CIII levels specifically in CM and VLDL1 compared to VLDL 2? This would greatly strengthen this point.
- For better interpretation of efflux studies, the authors may move table 3 in the beginning of section 3.3.
- The lipid parameters and efflux functions return to near baseline levels within 8h both in Ghigh and Glow Therefore in the discussion this detail should be added. One suggestions is under conclusion, line 442,
“Finally, our study underlines that healthy men subjects displaying fasting triglyceride levels within the normal range but exhibiting an undesirable transient postprandial TG response, share major common underlying mechanisms observed during metabolic hypertriglyceridemia, characterized by elevated fasting triglyceride levels and contributing to the development of atherosclerosis.
Author Response
Response to R2
Comment: This work investigate some of the functions of HDL in response to postprandial elevated triglyceride (TG) levels. The overall outcome is consistent with several previous studies on related topic. In this work the authors have done a systematic evaluation of some of the key functions of HDL and correlate it with TG levels. The study design is clearly explained and the data is satisfactory and supportive of their conclusion. These are some minor points that needs clarification for better understanding.
Response: We thank this Reviewer for his/her helpful and constructive comments which have allowed us to significantly improve the impact and the clarity of our manuscript
Comment: What is the reason for selectively choosing male subjects?
Response: Sex hormones influence expression of various key genes involved in the reverse cholesterol transport pathway such as CETP (PMID: 14729390), hepatic lipase (PMID: 10064731), or CLA1/ SR-BI (PMID: 17673517), thus modulating not only plasma lipoprotein profile but also functional properties of HDL particles (PMID: 18057374). To our opinion, a mixed population was therefore not appropriate in the context of the present study exploring the impact of postprandial triglyceridemia on reverse cholesterol transport efficacy, however it would be of interest to conduct similar explorations in a cohort of healthy non-dyslipidemic women.
Comment: In section 2.3, under study design, is it right to assume that after test meal, the subjects were not taking any other meal for 8h? If so, please add this detail in this section.
Response: Yes, it is correct, the subjects were not taking any other meal for 8h.
This point is now mentioned in the section 2.3.
Table 2 and 3, please give the time point that correspond to these values.
Response: Table 2 presents parameters determined on fasting samples obtained after an overnight fast, at 8:00 am.
This point in now clearly indicated on Table 2 footnote
Response: Table 3: Time 0h, indicates before meal intake, plasma levels and chemical composition of HDL subfractions were determined on samples obtained before meal intake at 11:30 am, and 2h, 4h, 6h and 8h after ingestion of the meal.
This point in now clearly indicated on Table 3 footnote
Comment: One of the conclusion from Table 2 and Figure 2 is elevated TG levels in CM and VLDL1 in Ghigh subgroup and the authors correlate this with elevated apoCIII levels. Do they have any data that show elevated apo CIII levels specifically in CM and VLDL1 compared to VLDL 2? This would greatly strengthen this point.
Response: ApoCIII levels in TRL subspecies are now presented in Figure 3 of the revised version of the manuscript and discussed as follows:
We presently observed significant higher circulating levels of CM-apoCIII, VLDL1-apoCIII and in a lesser extend of VLDL2-apoCIII in subjects from GHigh as compared to those from GLow (Figure 3AB). Moreover, VLDL2-apoCIII accounted for approximately 20% of total apoCIII bound to TRL particles (Figure 3C). These latter observations are in good agreement with earlier studies showing higher proportion of apoCIII bound to TRL particles and their remnants in hypertriglyceridemic subjects [27]. It is well established that apoCIII acts as an inhibitor of lipoprotein lipase and more specifically inhibits the LPL-mediated lipolysis of TG-rich lipoproteins [28]. ApoCIII equally impairs hepatic uptake of TRLs by remnant receptors including the LDL-R (Low density lipoprotein-receptor) and the LSR (lipolysis-stimulated receptor) [29]. Thus, undesirable postprandial hypertriglyceridemia presently observed in subjects from GHigh might primarily result from combination of increased apoCIII concentrations, reduced activities of LPL and hepatic remnant receptors, together with competition of intestinal and hepatic-derived lipoproteins for common removal pathways thus resulting in a delayed catabolism of TRLs and their remnants.
Comment: For better interpretation of efflux studies, the authors may move table 3 in the beginning of section 3.3.
Response: As suggested Table 3 has been moved in the beginning of section 3.3.
Comment: The lipid parameters and efflux functions return to near baseline levels within 8h both in Ghigh and Glow Therefore in the discussion this detail should be added. One suggestions is under conclusion, line 442,
“Finally, our study underlines that healthy men subjects displaying fasting triglyceride levels within the normal range but exhibiting an undesirable transient postprandial TG response, share major common underlying mechanisms observed during metabolic hypertriglyceridemia, characterized by elevated fasting triglyceride levels and contributing to the development of atherosclerosis.
Response: The following comment has been inserted on line 281, “To note that in subjects from both GHigh and GLow, plasma lipid parameters as well as circulating number of individual TRL subspecies return approximately to their respective baseline levels within 8h”.
The conclusion has been modified as suggested.
Round 2
Reviewer 1 Report
I have no further comment
Author Response
as requested, spell has been carefully checked